# Mastering Symbolic Operations: Augmenting Language Models with Compiled Neural Networks

## Abstract

Language models (LMs) proficiency in handling deterministic symbolic reasoning and rule-based tasks remains limited due to their dependency implicit learning on textual data. To enable fully rule comprehension ability, we explore how to incorporate compiled neural networks (CoNNs) which weight is specially designed into the architecture of LMs, to achieve high accuracy and robust performance. CoNNs are transformer-based neural networks that execute rules through artificially generated attention weights. Our method, which call "Neural Comprehension", by incorporating CoNN modules into the LM, the framework effectively tackles rule-intensive challenges. Our experiments on symbolic reasoning tasks and real-world arithmetic reasoning tasks demonstrate the superior performance of our method compared to existing techniques. Furthermore, our LM achieves flawless execution on symbolic operations tasks, highlighting the potential of our method in enabling LMs to possess true symbolic comprehension capabilities.

## 1 Introduction

Language models (LMs), particularly large language models (LLMs), have exhibited impressive performance on complex reasoning tasks [Brown et al., 2020, Zhang et al., 2022a, Chowdhery et al., 2022, Wei et al., 2022d,a, Suzgun et al., 2022]. Despite this, the proficiency of LMs in tackling deterministic symbolic reasoning and rule-based tasks is still limited [Welleck et al., Razeghi et al., 2022]. For example, GPT-3's arithmetic performance declines with higher digit numbers [Brown et al., 2020], and its mathematical accuracy is influenced by word frequency in training data [Razeghi et al., 2022]. Moreover, length generalization [Anil et al., 2022] remains a challenge even for 100-billion-parameter models, such as GPT-4 [Bubeck et al., 2023]. We hypothesize that these limitations stem from LMs' dependency on implicitly learning rules from textual data. During the training process, the primary objective of implicitly learning

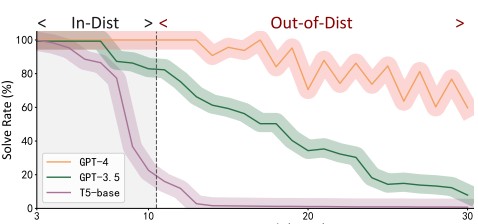

Figure 1: The length generalization of T5 (with fine-tune) [Raffel et al., 2020], GPT-3.5 (with few-shot) [Ouyang et al., 2022] and GPT-4 (with few-shot) on symbolic operations (Additional) tasks. The tasks included examples such as *"15673 + 3186"* (length = 10). To evaluate the model's proficiency, we conducted tests on tasks ranging from 3 to 30 digits, with longer than 10 digits being out-of-distribution of training data.

based on gradient Updating is to minimize the loss associated with the given textual dataset. As illustrated in Figure 1, a simple length generalization experiment using addition tasks with varying numbers of digits highlights this limitation. Performance deteriorates as test length increases, indicating that these models strongly rely on statistical patterns in the data rather than capturing fundamental logical structures. This reliance on implicit learning constrains LMs' accuracy in executing symbolic operations tasks. As a result, their performance suffers when confronted with out-of-distribution and rule-intensive tasks that require a more profound understanding of abstract rules.

Submitted to 37th Conference on Neural Information Processing Systems (NeurIPS 2023). Do not distribute.

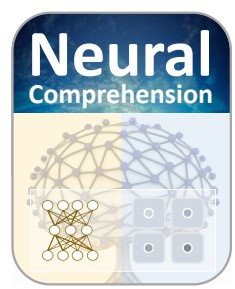

We propose a transformer-based language model framework, termed "Neural Comprehension", which synergistically integrates a pre-trained LM [Li et al., 2021b] and compiled neural networks (CoNNs) [Weiss et al., 2021] to achieve high accuracy and robust performance. CoNNs are neural networks but the rules are explicitly coded through transformer-liked structures and attention. Therefore CoNN is human-controllable, executing rules through artificially generated attention weights, and can achieve perfect accuracy once compiled network is done. Neural Comprehension relying solely on neural networks without requiring additional tools. It employs a token-by-token generation method, analogous to GPT-3, where each token can be generated by either the pre-trained LM or one of the CoNNs. We comprises a pre-trained LM and multiple sets of CoNNs. The implementation of the Neural Comprehension framework facilitates the integration of rule-intensive abilities and reasoning capabilities into LMs, endowing them with genuine symbolic comprehension skills.

In this work, we conduct extensive experiments to evaluate the performance of our proposed Neural Comprehension method on a variety of rule-intensive tasks. Our experimental results demonstrate the effectiveness of our approach in comparison with existing state-of-the-art techniques, such as vanilla fine-tuning, few-shot learning, and Chain-of-Thought reasoning. Specifically, Neural Comprehension outperforms these methods in terms of accuracy, efficiency, and interpretability, showcasing its superiority in handling rule-intensive tasks. Our study presents a strong case for the deployment of Neural Comprehension in language models, highlighting its potential to transform the landscape of symbolic reasoning and language understanding capabilities.

**Contributions** Our main contributions are as follows:

- We pioneer the development and implementation of flawless execution rule-intensive symbolic operations for language models that rely on neural networks. By employing a versatile and interpretable method, we successfully integrate CoNNs, which are explicitly coded and human-controllable, into the language model. Our method facilitates direct rule deduction without the need for learning from conditional probabilities, leading to a more robust and effective approach. (**Section** 3)

- To expand the application field, we leverage the In-context learning ability of large language models to auto generate CoNN. Our method can be easily extended to various symbolic operations tasks. (**Appendix C**)

- Our experimental results on controllable symbolic reasoning tasks and real-world numerical calculation tasks demonstrate the superior performance of our method in comparison to existing techniques. Notably, our language model achieves flawless execution on symbolic reasoning tasks. (**Section** 5.1 5.2 5.3)

- We also studied the potential of combining multiple CoNNs and found that adding correlated CoNNs can continuously increase performance, while adding uncorrelated CoNNs rarely leads to performance degradation. This provides a new approach for model fusion, enabling the model to easily acquire new knowledge. (**Section** 5.4)

## 2 Related Works

As model parameters, training calculations, and dataset sizes have increased, language models have gained new capabilities [Srivastava et al., 2022, Wei et al., 2022a], such as coding [Li et al., 2022b, Nijkamp et al., 2022], medical diagnosis [Li et al., 2021a, Xia et al., 2022], complex question-answering [Zhu et al., 2022, Daull et al., 2023], cross-language translation [Fan et al., 2021, Li et al., 2022a], few-shot learning [Brown et al., 2020, Perez et al., 2021], and thought chaining [Wei et al., 2022c, Weng et al., 2022]. However, these models also exhibit limitations as they generally learn superficial patterns rather than the innate logic and rules of language. Consequently, humans often find it challenging to trust the results provided by language models [Sarker et al., 2021, Moore, 2022].

**Pre-trained Language Models** encompass those trained on general-purpose corpora [Lewis et al., 2019, Scao et al., 2022] and specialized symbolic tasks [Geva et al., 2020, Lewkowycz et al., 2022]. They primarily aim to capture statistical patterns in language, which limits their capacity for symbolic

reasoning. Symbolic reasoning involves manipulating abstract symbols and logical rules to derive new knowledge [Shindo et al., 2021, Yang and Deng, 2021] and necessitates the ability to extrapolate to novel situations and reason about concepts absent in the training data [Fujisawa and Kanai, 2022]. Due to the constraints of gradient learning, neural networks face challenges in wholly solving symbolic reasoning problems.

**In-Context Learning** has emerged as a promising approach to address these challenges [Dong et al., 2022] and closely approximate the predictors computed by gradient descent [Akyürek et al., 2022]. By prompting the language model to generate an explanation before generating an answer, the chain of thought [Wei et al., 2022c, Kojima et al., 2022, Zhang et al., 2022b, Zhou et al., 2022a] encourages the model to think sequentially. This technique has been employed in various numerical and symbolic reasoning tasks, such as scratchpad prompting [Nye et al., 2021] for length generalization [Anil et al., 2022] and utilizing the chain of thought to perform arithmetic operations like summing pairs of single digits with carry [Zhou et al., 2022b]. However, this approach often necessitates substantial computational resources, and achieving perfect accuracy remains challenging.

**Augmented Language Models** have been proposed as an alternative, supplementing language models with external tools [Mialon et al., 2023]. Examples include generating Python code for numerical reasoning [Gao et al., 2022, Chen et al., 2022] or incorporating tool usage as a pre-training task [Schick et al., 2023]. However, using external tools lacks a unified framework with language models and instead relies on the normativity of program generation. Consequently, if a task demands higher-level abstraction or intricate and robust capabilities, such as Redefine [Wei et al., 2022b], Autoformalization [Wu et al., 2022], and Theorem Proving [Wu et al., 2020], the language model may struggle to solve it, even if it possesses the ability to operate external tools [Zhou et al., 2022b].

# 3 Methods

## 3.1 Preliminaries

**In-Context Learning (ICL)**, Recent studies on ICL algorithms have shown that the learning process of language models within the ICL framework is analogous to gradient descent [Akyürek et al., 2022]. Specifically, transformer-based in-context learners implicitly implement standard learning algorithms by encoding smaller models in their activations and updating these implicit models as new examples appear in the context. However, these models face challenges in rule-intensive questions, as the rules represent abstract, high-dimensional knowledge that cannot be directly learned from the data, resulting in difficulties with implicit learning.

**Compiled Neural Network (CoNN)**. The flexibility of neural networks to adjust their weights is a unique characteristic not found in the human brain. We propose incorporating CoNNs into LLM architectures to leverage this feature. The CoNN is a transformer-based neural network leveraging artificially compiled attention weights to execute rules. A transformer model comprises multiple attention layers and Multi-Layer Perceptron (MLP) layers. Each attention layer facilitates interactions between tokens, with the multiplication of query and key elements representing a "**Select**" operation in CoNN. Subsequent multiplication with value elements indicates an "**Aggregate**" operation. The MLP layer is responsible for the token itself and is referred to as the "**Zipmap**" operation [Weiss et al., 2021]. Utilizing the three operations (Select, Aggregate, and Zipmap) to represent the sequence-to-sequence process, we can convert this information into transformer weights [Lindner et al., 2023]. By stacking multiple attention layers, CoNN can address various human-defined rule understanding problems, such as mathematical calculations and symbol operations [1].

## 3.2 Neural Comprehension

Language models excel in language understanding tasks, while CoNNs achieve absolut accuracy in rule-intensive operation tasks using attention weights guided by abstract rules. To combine the language understanding capabilities of existing language models with accurate problem-solving for rule-based tasks (e.g., computation), we propose the Neural Comprehension, which integrates the language model's implicit learning parameters and CoNNs' explicit learning parameters. In Neural

---

[1]**Appendix B** provides a more detailed description of CoNN.

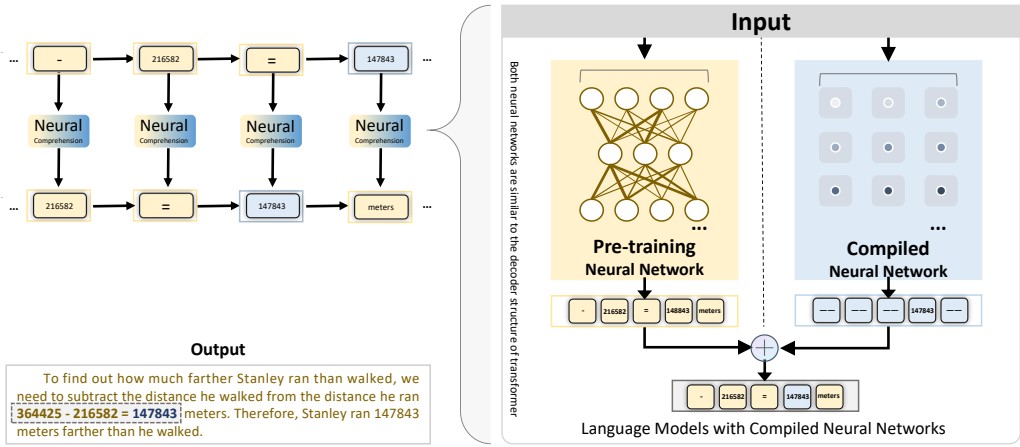

Figure 2: The architecture of Neural Comprehension.

Comprehension, CoNNs represent high-dimensional rules explicitly using multiple attention matrices and incorporate these with the original LM's attention matrix.

As illustrated in Figure 2, we maintain the use of a decoder architecture to iteratively generate the subsequent context step by step. In particular, the language model encodes the context and produces the textual and reasoning process context $D(x)$ step by step, while CoNNs handle sequence transformations involving rules. When a rule-required operation emerges, CoNN's attention is utilized to calculate specific values. The structure of Neural Comprehension is similar to MoE [Shazeer et al., 2017]. For example, when calculating *364425-216582*, the pre-trained language model output *148843*, which is incorrect. However, the `Subtraction` CoNN can correct the result to *147843* in the neural comprehension framework. This process encoded into context dynamically, improving intermediate results interpretability and final result accuracy.

Neural Comprehension combines LM and CoNNs in a piecewise function to perform gradient update. LLM hidden state output is $H_L = \left( H_{L_1} \cdots H_{L_{d_L}} \right)^\top \in \mathbb{R}^{d_L}, \quad H_{L_i} \in (0,1)$, and CoNN output is $H_C = \left( H_{C_1} \cdots H_{C_{d_C}} \right)^\top \in \mathbb{R}^{d_C}, \quad H_{C_i} \in (0,1)$ [2]. Specifically, we perform model fusion by adding the mapping from the last hidden layer representation to the vocabulary.

$$\hat{i} = \underset{i}{\mathrm{argmax}} \left[ \begin{pmatrix} I_{d_L}, 0 \\ 0, \beta I_{d_C} \end{pmatrix} \begin{pmatrix} H_L, 0 \\ 0, H_C \end{pmatrix} \right], \quad \beta \in \{0, 1\} \tag{1}$$

Within the Neural Comprehension, CoNNs manage sequence transformations involving rules. When the model encounters a rule-required operation, a gating mechanism determines whether to use CoNN's attention for computation. The gating mechanism assesses whether to maintain the initial output, provided by the pretrained language model, or modify it using the CoNN. where the model corrects the answer by applying a gradient to the in-context learning function through $\beta$. In Equation 1, since the hidden state output $H_{C_i}$ elements of CoNN are $\{0, 1\}$, when $\beta = 0$, the model adopts the original decoding token of LM. When encountering a rule calculation problem, $\beta = 1$, the model calculates the result by taking the maximum value of CoNN's hidden layer output $H_C$ and decodes the result from CoNN's vocabulary. Regarding the selection of $\beta$, since the CoNN involved in this paper is relatively simple, it is determined by the forward computation results of CoNN. For example, when we set up an `Addition` CoNN, we specify that the final result should be output when

---

[2]It is worth noting that $d_L$ and $d_C$ here refer to the vocabulary size of the Model's decode output. In this paper, for ease of implementation, the output vocabulary size of CoNNs' decode $d_C$ is generally less than 100 due to limitations in computing resources (detailed information is shown in **Appendix Table 1**). The Neural Comprehension combines the Pre-trained LM's hidden state output, $H_L$, and CoNN's output, $H_C$, using identity matrices $I_{d_L}$ (for $d_L$) and $I_{d_C}$ (for $d_C$) to concatenate them for model fusion.

167 encountering '=', so when encountering '=', $\beta = 1$. However, for larger-scale CoNN, we recommend
168 that a learnable gating network determine $\beta$.

### 3.3 Gradient Modification in Neural Comprehension

170 To better appreciate the benefits of our method in handling rule-intensive tasks and improving
171 accuracy, it is crucial to understand the gradient perspective of ICL. The optimization process in
172 ICL can be viewed as a search for suitable gradients to minimize the loss function. Due to the
173 implicit learning nature of standard ICL methods, gradients learned from data may not always be
174 ideal for addressing rule-intensive tasks. Therefore, our proposed method introduces an explicit
175 learning component to provide more appropriate gradient updates for such tasks, ultimately leading
176 to enhanced overall performance. In this section, we focus on elucidating the changes in the gradient
177 introduced by the Neural Comprehension model.

178 The gradient of the model during the execution of ICL can be partitioned into two categories based
179 on the origin of the gradients:

$$\text{Gradient} = \begin{cases} I_{d_1} & \text{Text} \\ I_{d_2} & \text{Rule} \end{cases} \tag{2}$$

180 Here, $I_{d_1}$ represents the gradients derived implicitly from the language model (LM) and corresponds
181 to the text-based learning aspect of the model. Conversely, $I_{d_2}$ represents the gradients explicitly
182 derived from the CoNNs, encoding rule-based knowledge. The Neural Comprehension model
183 integrates both gradient sources to optimize the ICL process.

184 In linear regression problems, the loss function can be expressed as a piecewise function according
185 to 1, here $P_1(x)$ is the LLM and $P_2(x)$ is CONN, the In-context-learner can be separate into two
186 process :

$$L = \left\| y - \beta^\top x \right\|^2 \tag{3}$$

$$= \begin{cases} \left\| y - \beta_1^\top x \right\|^2 & x \in P_1(x) \\ \left\| y - \beta_2^\top x \right\|^2 & x \in P_2(x) \end{cases} \tag{4}$$

187 Based on the partitioned gradient as defined in Equation 2, the overall gradient of the Neural
188 Comprehension model can be obtained by computing their individual gradients concerning the
189 respective $\beta$:

$$\underbrace{\frac{\partial L}{\partial \beta}}_{\text{Gradient}} = \begin{cases} \frac{\partial L}{\partial \beta_1} & x \in P_1(x) \\ \frac{\partial L}{\partial \beta_2} & x \in P_2(x) \end{cases} \tag{5}$$

190 This partitioning allows the Neural Comprehension model to specifically address the gradient require-
191 ments of both implicit learning via LM and explicit learning via CoNNs. It is crucial to note that
192 CoNNs are designed to minimize the loss associated with rule-based tasks, essentially providing an
193 optimal gradient for tasks involving rule-intensive operations. This leads to a substantial improvement
194 in the model's accuracy for rule-based tasks, as the gradient updates provided by CoNNs are more
195 suitable for rule learning compared to the initially available gradients from the LM. By amalgamating
196 the both of gradient sources, the Neural Comprehension model achieves a more refined optimization
197 of in-context learning. Additionally, from the perspective of gradients, our approach surpasses
198 conventional data-driven implicit learning techniques as it integrates explicit rule-based learning
199 mechanisms that exhibit more suitable gradient updates for rule-intensive questions. The Neural
200 Comprehension model effectively balances the need for implicit and explicit learning within the ICL
201 framework, leading to an enhanced overall performance in terms of accuracy and interpretability.

## 4 Experimental Settings

203 In this study, we primarily explore the capacity of language models to address symbolic reason-
204 ing tasks, concentrating on three areas: symbolic operations, symbolic reasoning, and arithmetic
205 reasoning.

206 **Symbolic Operations**    Building upon the approaches developed by Anil et al. [2022] and Qian
207 et al. [2022], we examine the following tasks: Parity, Reverse, Addition and Subtraction. These
208 tasks do not require complex text understanding, but only require faithfully implementing symbolic
209 operations and outputting the corresponding results.

210 **Symbolic Reasoning**    We employ the experimental framework of Wei et al. [2022c] for the two
211 tasks, Last Letter Concatenation and Coin Flip. These tasks require a combination of language
212 understanding and rule comprehension abilities.

213 **Arithmetic Reasoning**    To evaluate the method's generalization ability from symbolic operations
214 to arithmetic reasoning in addition and subtraction tasks, we use five established arithmetic reasoning
215 datasets: AddSub [Hosseini et al., 2014], SingleEq [Koncel-Kedziorski et al., 2015], MultiArith [Roy
216 and Roth, 2016], GSM8K [Cobbe et al., 2021], and SVAMP [Arkil et al., 2021]. Additionally, we
217 introduce the AddSub$^+$ dataset, containing tasks of varying complexity based on the number of digits
218 involved in arithmetic operations, ranging from 1-digit addition to 20-digit addition/subtraction tasks.

## 219 5   Ecperiment and Result

### 220 5.1   Symbolic Tasks

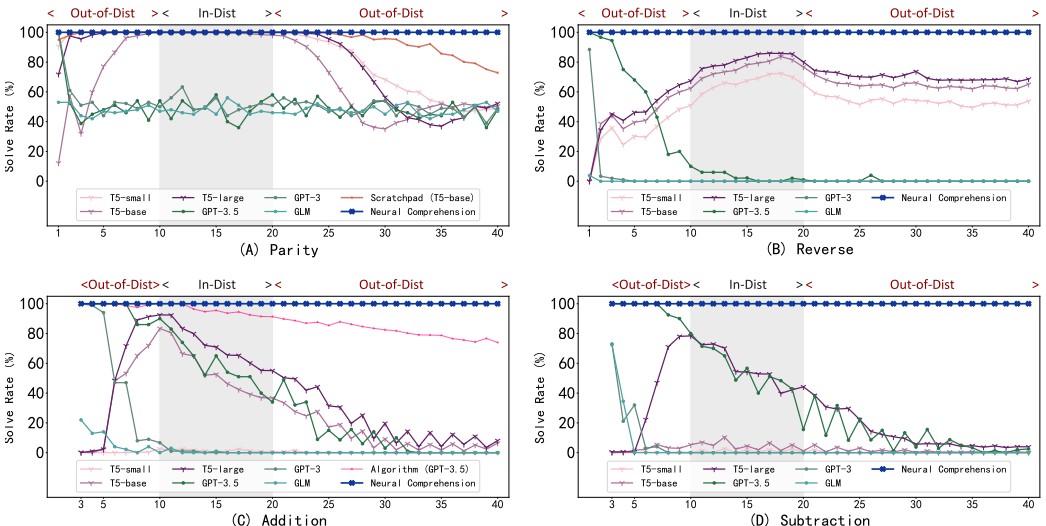

Figure 3: Comparison of Neural Comprehension and other implicit learning-based methods in symbolic operations tasks to test length generalization performance. In this, the T5 model uses the Vanilla Fine-tune method for learning, and LLMs use the Few-shot learning method. In Neural Comprehension, each task has a different CoNN, namely `Parity`, `Reverse`, `Addition`, and `Subtraction`.

| Techniques | In-distribution | Out-of-distribution | Time and Space Complexity | Interpretability |
|---|---|---|---|---|
| Vanilla Fine-tune (For LM) | ✓✓ | ✗ | ✓✓ | ✗ |
| Vanilla Few-shot (For LLM) | ✓ | ✓ | ✓✓ | ✗ |
| Scratchpad [Anil et al., 2022] | ✓✓ | ✓ | ✗ | ✓ |
| Algorithmic [Zhou et al., 2022b] | ✓✓ | ✓ | ✗ | ✓ |
| **Neural Comprehension (Ours)** | ✓✓ | ✓✓ | ✓✓ | ✓✓ |

Table 1: Performance on Symbolic operations tasks of five techniques that language models admit: (1) Vanilla Finetuning, (2) Vanilla Few-shot, (3) Scratchpad (Chain-of-Thought reasoning), (4) Algorithmic (Chain-of-Thought reasoning) and (5) Neural Comprehension. We find that the first four learning-based methods have different modes of failure regarding in and out-of-distribution coverage for symbolic operations. However, Neural Comprehension has strong advantages in terms of length generalization, efficiency, and interpretability. ✗ signifies poor ✓ signifies nontrivial, ✓✓ signifies near-perfect performance. (*) Refers to task-dependency.

221 In this study, we conduct a length generalization experiment [Anil et al., 2022] to examine the
222 distinctions between the Neural Comprehension and learning-based methods, as depicted in Figure 3.
223 Our experimental design encompasses $1000 \times 40$ independent test sets, comprising problems with

varying digit lengths from 1 to 40 digits. 10 to 20 digits within the range are provided by us for methods based on implicit learning for training; during the testing phase, this range is called In-Dist. Furthermore, we present results for both Scratchpad [Anil et al., 2022] and Algorithmic [Zhou et al., 2022b] approaches.

The results of our experiment demonstrate that the Vanilla Fine-tune (red lines) method performs optimally on the in-domain (10-20 digit) training set, while its performance deteriorates for both more simplistic and more intricate. This finding suggests that the absence of relevant samples in the training set may cause gradient descent-based language models to underperform on both simpler and more complex tasks. As further discussed in the **appendix D.1**, this phenomenon can be attributed to the inherent generalization limitations of statistical models and the position bias of language models.

Considering the Vanilla Few-shot method (green lines), we determine that its performance is less impacted by the prompt sample range compared to Vanilla Fine-tune. Large language models, which are trained on extensive text corpora, excel at solving more straightforward problems such as symbolic operations within a ten-digit range. Nevertheless, performance remains below par for test sets with more than ten digits, even when prompted with 10-20 digit samples.

Observing CoT-like methods (we use GPT-3.5), including Scratchpad and Algorithmic, unveils their robust length generalization capabilities. Scratchpad works by requiring large language models to record intermediate steps, while Algorithmic employs a similar approach to record the carry operations involved in the addition process. This can be primarily attributed to their proficiency in decomposing complex problems into smaller incremental steps and maintaining intermediate states. However, these methods necessitate substantial computational resources, and extending the length beyond the input limit of the model becomes challenging.

Our study reveals that Neural Comprehension attains remarkably high accuracy in symbolic operations. This implies that Neural Comprehension, unlike conventional methods, does not rely on training data and remains unaffected by discrepancies in input lengths for in-distribution and out-of-distribution data. Consequently, it alleviates the requirement for step-by-step work tracking, and language models with CoNNs only need relatively fewer computational steps to execute sequence operations directly. Encoding rules into neural network modules endows us with greater interpretability, enabling language models to flawlessly perform purely symbolic operation tasks.

## 5.2 Symbolic Reasoning

In this section, we investigate the performance of Neural Comprehension in terms of symbolic reasoning capabilities. Our hypothesis is that, although pretrained Language Models (LMs) demonstrate strong language understanding abilities, they lack the capacity to deduce and comprehend rules regarding symbolic reasoning tasks. Thus, we aim to evaluate whether the incorporation of compiled neural networks in the form of CoNNs can address this limitation and improve the LM's symbolic reasoning abilities.

To assess the performance of the rule comprehension component (CoNNs) in symbolic reasoning, we devise an experiment that measures the model's accuracy using intermediate processes and represents them in a "Chain of Thought"-like manner. In doing so, the experiment decomposes language understanding and rule comprehension explicitly into simpler outputs, avoiding the complexities of reasoning and additional error propagation in the models. Example outputs from this approach can be found in **Appendix F**. We observed that neural com-

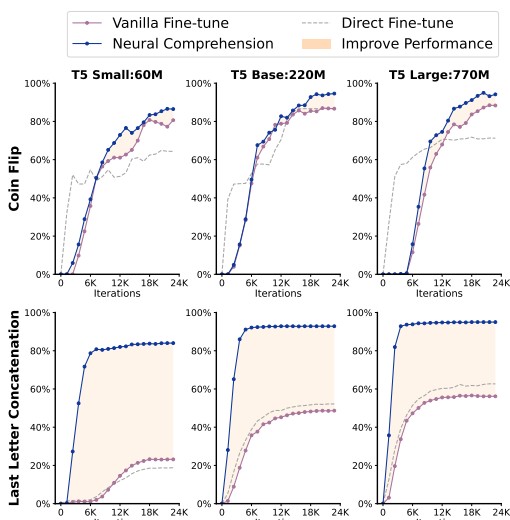

Figure 4: In the iterative process of gradient descent during training. The bleu line represents a language model that incorporates neural comprehension, and the red line represents the original language model. Additionally, we provide Direct, which is a direct prediction of the final result, as a reference.

prehension improves the symbolic reasoning capabilities of pre-trained language models in most cases (Neural Comprehension almost always outperforms Vanilla Fine-tune in Figure 4), and can fit faster. This observation suggests that the introduction of compiled neural networks has a positive impact on pretrained LMs, addressing rule comprehension limitations in symbolic reasoning tasks.

## 5.3 Arithmetic Reasoning

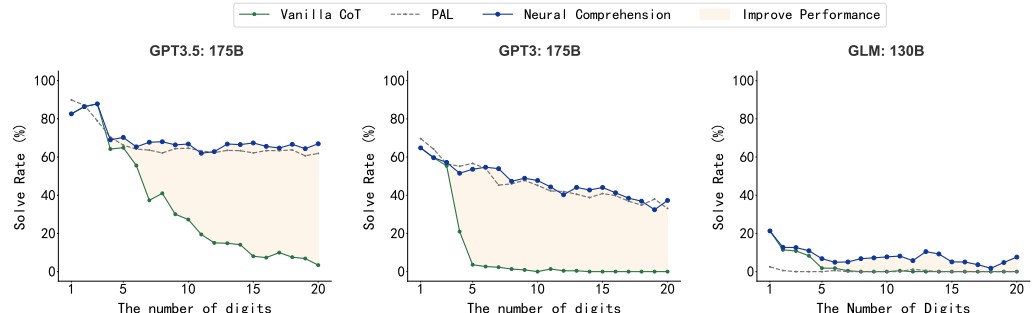

Figure 5: We conducted simulations of the AddSub dataset with varying digits by modifying the "lEquations" parameter. We then tested the performance of three LLMs with and without Neural Comprehension in generating CoT outputs for AddSub$^+$. And we reported the solve rates of three LLMs and compared the solve rates of using additional tools (PAL [Gao et al., 2022]).

Arithmetic reasoning serves as a suitable testbed for evaluating language models and their ability to address real-world problems. In this study, we examine the AddSub$^+$ dataset variants that involve different digit lengths, utilizing the `Addition` and `Subtraction` models from the CoNNs family. Notably, the capabilities of Neural Comprehension extend beyond these tasks, as CoNNs can also simulate calculators that support multiplication and division operations, and potentially perform linear algebra computations or even in-context learning algorithms that employ backpropagation [Giannou et al., 2023].

To evaluate the impact of Neural Comprehension on arithmetic reasoning, we compare the output of vanilla CoT language models and those incorporating Neural Comprehension, using the vanilla CoT baseline as a reference. As demonstrated in Figure 5, the vanilla CoT model struggles to extrapolate and solve arithmetic problems involving longer digit lengths. However, integrating Neural Comprehension significantly improves the performance of language models on such complex arithmetic tasks. Since we only incorporated the `Addition` and `Subtraction` CoNNs, we attribute the observed performance enhancement to the increased computational accuracy of the language model. For further evidence, we present additional experimental results on widely-used arithmetic reasoning datasets in **Appendix D.2**, which reinforce the benefits of using Neural Comprehension over the vanilla CoT model.

In comparison to language models employing external tools like PAL [Gao et al., 2022], our findings suggest that generating accurate code for the less code-trained GLM-130B model might be challenging for PAL, resulting in performance levels inferior to those of the vanilla CoT. This outcome indicates that language models offer greater flexibility, whereas external tools may have difficulties in more complex or unique situations. The integration of compiled neural networks appears to be a more promising approach, as evidenced by the performance improvements observed in our experiments.

Specifically, when language models encounter intricate arithmetic tasks that involve nested operations or multi-step calculations, the integrated CoNNs can efficiently handle these operations, allowing the language model to focus on higher-level reasoning. In contrast, the use of external tools often requires explicit coding and may not generalize effectively to more complicated scenarios. In conclusion, our results demonstrate that incorporating compiled neural networks into language models provides a more robust and versatile solution for arithmetic reasoning and related challenges, underlining the superiority of this approach over external tools such as PAL.

## 5.4 Ablation and Analyses: Module Combination for Neural Comprehension

Efficiently deploying multiple CoNNs is crucial for achieving exceptional Neural Comprehension performance. As depicted in Figure 4, the amalgamation of distinct CoNNs, tailored for both symbolic

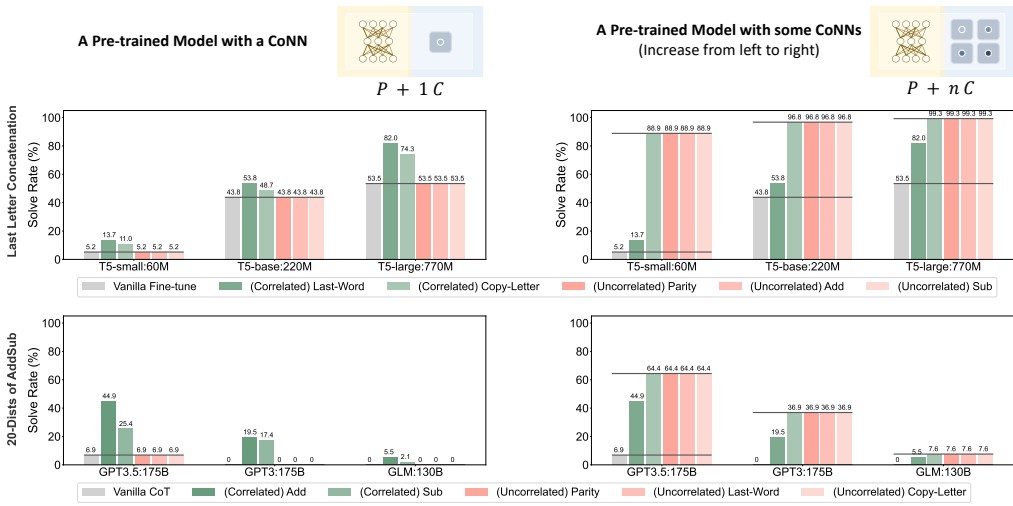

Figure 6: In Neural Comprehension framework, the performance of multiple different module combination is demonstrated. The left side shows the effect of combining a pre-trained language model with a CoNN, while the right side shows the impact of combining a language model with multiple CoNNs. For different tasks, we categorize CoNNs as Correlated (green) and Uncorrelated (red), indicating whether the CoNN is related to the current task or not.

and arithmetic reasoning tasks within the language model framework, can lead to remarkable benefits. It is observed that integrating pertinent CoNNs bolsters the performance of the initial language model, whereas the inclusion of unrelated language models rarely causes detrimental effects, regardless of whether single or multiple CoNNs are combined.

This can be ascribed to the refined design of the Neural Comprehension framework, which ensures the precise execution of assigned tasks by CoNNs without interference from irrelevant modules. Each CoNN module is adept at generating the appropriate output when needed, thereby preventing the emergence of erroneous results from unrelated components. Importantly, as seen in **Appendix B.3**, the parameter count for each CoNN module ranges from 1/1000 to 1/1000000 of that for GPT-3, and the experiments in **Appendix D.3** show that the inference latency in the neural understanding framework only increases by 1%-3% compared to Vanilla.

This observation underscores the remarkable scalability of the Neural Comprehension framework, which possesses the capability to not only accommodate existing knowledge concepts but also assimilate novel ones as the number of CoNNs expands. Theoretically, the integration of tens of thousands of CoNN modules within language models holds the potential to foster a comprehensive understanding of concepts.

# 6 Conclusion

We have observed that pretrained language models lack an intrinsic comprehension of rule-based concepts and explored how Neural Comprehension can integrate compiled neural networks into the language model framework in a simple and generic manner. We demonstrated the superiority of our approach over existing learning-based method, Without external tools, our approach enables language models to perform nearly perfect symbolic operations and can be applied to more realistic arithmetic reasoning tasks.

Our study opens new avenues for language models, such as the investigation of more complex CoNNs related to higher-order abstract reasoning, the development of more advanced gating mechanisms for smoother integration, and the exploration of other domains in which Neural Comprehension could exhibit significant advantages. Furthermore, our framework provides a foundation for future work on unifying both implicit and explicit learning in language models and facilitating the seamless.

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
