# OpenReview forum: "Mastering Symbolic Operations: Augmenting Language Models with Compiled Neural Networks"
_NeurIPS.cc/2023/Conference — Submitted to NeurIPS 2023_

### Official Review · Reviewer_gBrw · 2023-06-14

**Soundness:** 3 good
**Presentation:** 2 fair
**Contribution:** 2 fair
**Rating:** 3
**Confidence:** 3

**Summary:**

Large language models lack expertise skills and this is reflected in their limited capability for arithmetic etc.

The paper proposes a method to integrate a CoNN (compliled neural networks) into an LLM via gating. Such integrations allow better performance for rule intensive tasks such as symbolic reasoning, arithmetic reasoning etc. To perform this, the paper proposes a gating mechanism although the implementation seems rule based triggering (line 167).

With the proposed mechanisms, the authors evaluate on arithmetic tasks (5.1) where the model achieves consistently 100% accuracy. On symbolic reasoning (5.2) and arithmetic reasoning, the method also performs better than finetuning appraoches in terms of performance or data efficiency.







**Strengths:**

The paper correctly identifies the limitations of LLMs and propose a novel approach to tackle the problem. The solution consists of a neural machine that serves as expert for symbolic/arithmetic operations.

Empirically, the paper demonstrates strong results over competitive baselines on arithmetic/symbolic reasoning approaches.

**Weaknesses:**

The novelty introduced in the paper doesn't quality a Neurips paper. The novelty of the paper concretely is using the rule based trigger to combine a CoNN with an LLM; neither components consist of the paper novelty.

There are various presentation issues that make the paper quite hard to follow:
- The second contribution item is put in Appendix, I read it but please note that even reviewers are not obliged to read those materials in Appendix to judge the paper. Related to this remark, CoNN is only introduced and referenced, for people not familiar with the technology, there is nowhere in the paper to know how it works.
- Section 3.2 introduces the gating between LLM and CoNN, I think equation (1) has a flaw since it involves choose argmax from a matrix which seems ill defined and I think the authors mean to concatenate HL and HC instead. The gradient flow (3.3) doesn't give further insight rather than it is a gating mechanism.
- I am confused by the illustrative figures in the paper. Figure 2 has the input text on top which is easily confused to a paper title and I feel Figure 1 is not relevant to the paper.



**Questions:**

For experiments other than 5.1, the model doesn't achieve 100% accuracy (even for tasks like the last letter). How do authors explain this difference please? Is it because some arithemetic CoNN has been implemented while the last letter one has to be learnt?

Line 162, when beta equals 1, left side of equation one is an identity matrix, so the model seems to choose between HL and HC max, but the authors say that it will take the HC max, why?

**Limitations:**

Unless Neural Comprehension machines are widely used, I don't see why this approach is particularly useful: I don't see the advantage compared to API calling approach adopted by today's industry. Particularly, as shown in this paper, many operations have to be implemented individually (subtraction, addition, etc.)

---

> ### Author Rebuttal · Authors · 2023-08-09
>
> Thank you for taking the time to review our submission and providing constructive feedback. Below are our rebuttals to address your concerns. We hope this helps clarify any misunderstandings.
>
> ---
>
> **Review:** The novelty of the paper concretely is using the rule based trigger to combine a CoNN with an LLM; neither components consist of the paper novelty.
>
> **Response:** Our key contributions are: 1) A novel integration of CoNNs and LMs using a gating mechanism that combines their complementary strengths in a plug-and-play manner. And to the best of our knowledge, no prior work has approached the combination of these two in the same way. 2) We propose AutoCoNN to automatically generate CoNNs using LLMs. This enables applying our framework to rule tasks without manual effort. We will add more implementation details of CoNNs and AutoCoNN to the main text to strengthen novelty.
>
> ---
>
>
> **Review:** CoNN is only introduced and referenced, for people not familiar with the technology, there is nowhere in the paper to know how it works.
>
> **Response:** Thank you for the feedback on supplemental material and CoNN details. We will move the core AutoCoNN content into the main text and provide more specifics on CoNNs, as they form a key contribution.
>
> ---
>
> **Review:** (1) A flaw since it involves choosing argmax from a matrix which seems ill-defined and I think the authors mean to concatenate HL and HC instead. The gradient flow (3.3) doesn't give further insight rather than it is a gating mechanism.
>
> **Response:** The vocabulary of CoNN is obtained by expanding the original vocabulary of the LM with a new vocabulary. Therefore, in Equation 1, the hidden state representation matrix is obtained by extending the original hidden state representation matrix of the LM with the hidden state matrix on the RASP vocabulary through a block matrix operation, resulting in a larger matrix. The selection is then performed using \beta. Thank you for pointing this out; we will modify the equation to correctly represent our approach for clarity. Regarding the gradient flow (3.3), we presented it to illustrate how our method exploits the gradient updates from both the LM (implicit learning) and CoNNs (explicit learning).
>
> ---
>
> **Review:** Figure 2 has the input text on top which is easily confused with a paper title and I feel Figure 1 is not relevant to the paper.
>
> **Response:**  We'll adjust the layout, the new Figure 2 can be viewed in the PDF we submitted. Figure 1 was meant to illustrate the concept of so-called "length generalization" in language modeling, establishing the motivation for our research.
>
> ---
>
> **Review:** The model doesn't achieve 100% accuracy (even for tasks like the last letter). How do authors explain this difference please?
>
> **Response:** For all reasoning tasks (including symbolic reasoning and arithmetic reasoning), solving them requires the LM to have language understanding and rule comprehension (line 272). For example, the last letter task "Take the last letters of the words in Apple Pencil and concatenate them." needs to first sequentially judge 'Apple' and 'Pencil' from it, and then CoNN will output the last letters "e" and "l" for these words. For arithmetic reasoning, it relies on the LM to understand the problem and list the formula so that the CoNN can output the result.
>
>
> ---
>
> **Review:** Line 162, when beta equals 1, left side of equation one is an identity matrix, so the model seems to choose between HL and HC max, but the authors say that it will take the HC max, why?
>
> **Response:**  The reason is that when \beta=1, H_L is the hidden state output after passing through the activation function, for example [0.23,0.84,...,0.02]. The H_C is the hidden state output of CoNN. Since CoNN is a transformer model converted from rules, its output has and only has one value of 1, with the rest being 0, for example [0,1,...,0]. Therefore, according to the calculation of Formula 1, the value is always the max of H_C.
>
> ---
>
> **Review:** I don't see the advantage compared to API calling approach adopted by today's industry.
>
> **Response:** Thank you for this observation. Indeed, our NC (Neural Comprehension) methodology addresses certain limitations inherent in traditional API calling approach.
>
> Firstly, NC framework, with its capacity for flawless execution of rule-intensive tasks, present an attractive alternative where accuracy is critical. As shown in Figure 5 (Section 5.3 Arithmetic Reasoning), the performance of NC is not inferior to methods calling APIs (such as PAL [1], which utilizes LLMs to generate Python code instead of directly getting the answer). We excerpt some of the contents in Figure 5 into the following table:
>
> | Method | 1-dist AddSub | 5-dist AddSub | 10-dist AddSub | 15-dist AddSub | 20-dist AddSub |
> |-|-|-|-|-|-|
> | GPT-3.5+CoT | 81.7 | 65.3 | 28.6 | 8.9 | 3.2 |
> | GPT-3.5+PAL | **89.7** | 66.2 | 66.4 | 63.5 | 63.1 |
> | GPT-3.5+NC | 81.9 | **70.2** | **67.5** | **67.8** | **67.4** |
> |-|-|-|-|-|-|
> | GLM-130B+CoT | **21.2** | 1.3 | 0.0 | 0.0 | 0.0 |
> | GLM-130B+PAL | 3.4 | 0.1 | 0.0 | 0.0 | 0.0 |
> | GLM-130B+NC | **21.2** | **7.8** | **8.2** | **5.3** | **8.1** |
>
> We can find that for high-performance LMs like GPT-3.5, relying solely on the NC by transformer architecture is far superior to normal CoT methods, and is not inferior to methods using APIs like PAL. On the other hand, for weaker LMs like GLM-130B (in fact, including open-source LLMs like Llama and all smaller LMs belong to this category), due to the lack of ability to call APIs, PAL is rendered useless, but NC can still improve its performance.
>
> ---
>
> We hope this information can help resolve your confusion and re-evaluate the rating of our paper. At the same time, we will definitely further improve this paper based on your useful suggestions to make it more clear.
>
> [1]Gao L, Madaan A, Zhou S, et al. Pal: Program-aided language models[C] International Conference on Machine Learning. PMLR, 2023: 10764-10799.

---

> > ### Comment · Reviewer_gBrw · 2023-08-15
> > **Thank you for the feedback**
> >
> > Thank you for the detailed feedback.
> >
> > Through the rebuttal, the novelty is further clarified: A novel integration of CoNNs and LMs using a gating mechanism that combines their complementary strengths in a plug-and-play manner. This is indeed a contribution that I see in the paper but didn't write clearly in my first review. I thank the authors to clarify about the math technical details as well as discussing thoroughly about the presentation issues for which I think they will improve.
> >
> > I also read the discussions with other reviewers and it is helpful thanks to all. My main concern is the scalability of the approach (aik1) and how they generalize to new tasks. The AutoCoNN seems promising inside the discussion but at the moment, it is part of the appendix. The rule based design doesn't impact the main novelty, and I suggest the authors stick to its simplest form instead of introducing/discussing the beta that is not implemtend again, this is part of the presentation issues that many reviewers have raised.
> >
> > Meanwhile, I still have the confusion about whether this approaches offer any theoretical advantages over API based approaches, since both of them can perform execution at 100% accuracy. The empirical part only shows that integrating CoNN improves on small models which seems a little bit obvious in the sense that if your module is neural then you can always integrate it and on smaller models the integrated model should perform better. I might be wrong in this case, but personally I fail to see the advantage of making many things in the form of CoNN and that impacted my decision about accepting the paper. In fact, I think there might be tasks that can illustrate its benefits, but the examples tested in the paper can all be addressed by relatively code snippets where I don't see why we should use CoNN in this case.
> >
> > I will maintain my score and my decision.

---

> > > ### Author Response · Authors · 2023-08-17
> > > **Re: The scalability of the approach and how they generalize to new tasks**
> > >
> > > Thank you for your time in reviewing our paper and providing us with valuable feedback. Allow us to address your concerns further.
> > >
> > > ---
> > >
> > > **Review:**  The AutoCoNN seems promising inside the discussion but at the moment, it is part of the appendix.
> > >
> > > **Response:** When writing the paper, we placed AutoCoNN in the appendix since it is considered a toolkit. In fact, the Parity Model, Reverse Model, Last Letter Model, and Copy Model mentioned in the experiments of this paper are all constructed by AutoCoNN. To make the presentation more complete and better highlight the importance and novelty of this aspect, we will include further AutoCoNN discussions in the main text.
> > >
> > > ---
> > >
> > > **Review:** I suggest the authors stick to its simplest form instead of introducing/discussing the beta that is not implemented again.
> > >
> > > **Response:** We appreciate your suggestion. The details of the gating mechanism implementation will be clarified in the revised manuscript. We will simplify the beta that is not implemented in the revised manuscript without losing clarity and effectiveness.

---

> > > ### Author Response · Authors · 2023-08-17
> > > **Re: The advantages of Neural Comprehension**
> > >
> > > We believe our approach with Neural Comprehension (NC) provides a novel and powerful alternative. Let's address the benefits of this approach considering two scenarios:
> > >
> > > ---
> > >
> > > **1) Scenarios supporting APIs and language models have function calling capabilities**: Firstly, it is important to highlight that Neural Comprehension (NC) relies solely on the original transformer architecture and does not necessitate an additional processing step or external tools. This directness is an inherent advantage over typical API-based approaches that need to generate code snippets step-by-step, receive feedback, and finally get an answer based on the feedback. This indirect process is prone to accumulating errors and reduces efficiency.
> > >
> > > For example, consider using GPT-3.5 with API function calling capabilities to solve an arithmetic reasoning task. And whenever the LM generates an <EOS>, its generation will be stopped.
> > >
> > > > Q:  iWatch shows that Stanley runs an average of 364,425 meters and walks 216,582 meters per month. If he continues at this frequency, how much farther will Stanley have run than walked after one year?
> > >
> > > **API:**
> > > >  `<FUNCTION>{'name':'CALL_PYTHON','arguments':'def sub():\n    return 364425-216582\nsub()'}</FUNCTION>[EOS]<RETURN>{'name':'CALL_PYTHON','content':'147843'}</RETURN><EOS>`A: Firstly, we subtract the walking distance from the running distance to find out how much farther Stanley runs than walks in a month: 364425 meters - 216582 meters = 147843 meters <EOS>`<FUNCTION>{'name':'CALL_PYTHON','arguments':'def multiplication():\n    return 147843 * 12\nmultiplication()'}</FUNCTION>[EOS]<RETURN>{'name':'CALL_PYTHON','content':'1774116'}</RETURN><EOS>`Over a year, he will run 147843 meters/months * 12 months = 1774116 meters <EOS>
> > >
> > > **NC:**
> > > > A: Firstly, we subtract the walking distance from the running distance to find out how much farther Stanley runs than walks in a month: 364425 meters - 216582 meters = *147843* meters. Over a year, he will run 147843 meters/months * 12 months = *1774116* meters <EOS>
> > >
> > > An API-based approach would typically involve generating Python code to do arithmetic calculations, running the code using an external Python interpreter, and then providing the answer based on the interpreter's output. Frequent API calls can also result in an excessive amount of irrelevant content in the context of language models. By contrast, a compiled neural network for arithmetic in our NC framework would handle this operation internally and directly output the resultant (The italicized text in the example.). On the other hand, for more complex or unique situations, generating correct API-call codes, especially for less code-trained LMs, might be challenging. In fact in the above table, PAL's application on GLM-130B significantly underperformed the NC, which directly calculates the answer. And the GPT-3.5+NC is not inferior to GPT-3.5+PAL due to the fact that NC avoids the occurrence of incorrect code generated by PAL.
> > >
> > > Furthermore, building neural modules (CoNNs) into LMs ensures that our model remains fully differentiable, which is not the case with approaches that call APIs. This differentiability is crucial when fine-tuning the entire model to adapt to new data or tasks. If we want to revise the weight based on feedback (for instance, reinforcement learning from human feedback or other settings with sparse delayed rewards), API-based approaches are not differentiable and thus cannot handle such tasks efficiently.
> > >
> > > **2) Scenarios without API support or language models lacking function calling capabilitie.** NC is applicable regardless of the specific LM's ability to call functions, making it advantageous in scenarios where APIs cannot be leveraged or the language model lacks effective function calling capabilities.
> > >
> > > As stated in line 190, CoNN essentially provides the optimal predefined neural network for a specific symbolic operation task. Thus, when combined via the gating mechanism, we can ensure that the overall network will perform no worse than any network obtained via gradient descent, leading to a model with fewer parameters, lower computational cost, and superior symbolic operation capabilities.
> > >
> > > ---
> > >
> > > We sincerely appreciate your valuable feedback. We will incorporate the advantages of NC compared to API into the revised version of the paper to enhance its quality and arguments.

---

> > > ### Author Response · Authors · 2023-08-19
> > > **Re: The advantages of Neural Comprehension - 2**
> > >
> > > We summarize these advantages as **Efficiency**, **Unified End-to-End Neural Network Framework**, and **High Applicability**.
> > >
> > > - **Efficiency:** NC eliminates the need for generating additional code and feedback during the text generation process. This results in more efficient operation of the language model and reduces the computational resources required for generating extra code.
> > >
> > > - **Unified End-to-End Neural Network Framework:** NC forms a complete neural network that does not require the involvement of an external interpreter. Thus, the model remains fully differentiable and trainable within a causal language model training framework. The reason for using APIs in the past was because language models were unable to achieve complete accuracy [1]. Thus, APIs were indirectly invoked as a way to reduce their hallucinations, but this might also compromise efficiency and performance.
> > >
> > > - **High Applicability:** API-based methods can only be used in large language models that have undergone additional code training. Our original paper's experiments have shown that even models like GLM-130B struggle to effectively utilize API methods (as reflected in our GLM-130B+PAL test results). However, our proposed NC framework has demonstrated the ability to significantly enhance performance across a variety of model scales, from small scale models like T5-Small (60M) to larger models such as GLM-130B (130B) and GPT-3 (175B).
> > >
> > > We hope these explanations address your concerns. If you have any further questions before the deadline, we would be more than happy to continue this dialogue.
> > >
> > >
> > > ---
> > >
> > > [1] Mialon G, Dessì R, Lomeli M, et al. Augmented language models: a survey[J]. arXiv preprint arXiv:2302.07842, 2023.

---

### Official Review · Reviewer_QFXM · 2023-07-04

**Soundness:** 2 fair
**Presentation:** 3 good
**Contribution:** 3 good
**Rating:** 6
**Confidence:** 3

**Summary:**

Authors have proposed a novel way to augment large language model called Neural Comprehension to improve symbolic reasoning in tasks where rule-based execution is required by design such as numbers summation. The core idea behind their method is to augment the LM with compiled NN (CoNN) for a specific task is a manner of mixture of experts where they design a policy which determines LM or CoNN will be executing the next token prediction at each time step. In addition, they described how their method could be used with in-context learning (ICL). Authors perform set of experiments in symbolic operations (parity, reverse. addition, subtraction), symbolic reasoning (concat, coin flip) and arithmetic reasoning. They show empirically how their method outperforms stand-alone LMs finetuned on corresponding tasks data. Finally they show a potential of combining multiple CoNN with the given LM to increase the task capability of the final Neural Comprehension model.

**Strengths:**

This work proposes an original way to combine LMs with CoNN networks and analyze the performance using multiple correlated or uncorrelated CoNNs together.
The wide range of symbolic experimental tasks show that authors performed high quality experimental investigation.

**Weaknesses:**

I think major weaknesses of this work is (1) a hardcoded structure of CoNNs under consideration and (2) hard coded policy of choosing the LM vs CoNN component by connecting that to task-based properties. I believe the most interesting part would be to learn the beta factor which also seems to be very challenging.

Authors claimed that their work suggests potential improvements from using Neural Comprehension in other tasks, but they did not mention how to get CoNNs for these tasks? In general, the discussion about CoNN design and implementation is somewhat skipped in the paper while it seem to be a crucial factor in this paper's impact.

**Questions:**

There are some grammar and syntax typos in the paper, even in the section titles, please fix that.

A clarification question: from my understanding you have done some hard coding of the beta policy such that i.e. it only triggers ICL gradient when it sees "=" token. Does this mean that this approach will not work at all if you use sequence "4 + 4 equals 8"? If so, do you have any ideas in mind how to make it work? In general I like this idea of plug-in CoNNs, but they need to be more seamless and fluent without requiring such hard coding in the task level.

**Limitations:**

Authors discuss some statistical limitation aspects of LMs in the appendix and how their proposed method alleviates that. However, I did not find explicit discussion of limitations of their own approach except for describing future work.

---

> ### Author Rebuttal · Authors · 2023-08-09
>
> We appreciate your recognition of our work's novelty and significance in improving symbolic reasoning tasks.
>
> ---
>
> **Review:** Hardcoded structure of CoNNs under consideration
>
> **Response:** You correctly pointed out that in the current implementation, \beta and CoNNs are hardcoded and manually specified. We acknowledge this limitation and mentioned it in the Limitations and Future Work section. However, the primary objective of this research was to introduce the concept of incorporating CoNNs into the LM's architecture to enhance rule comprehension capabilities. The fixed \beta avoids introducing additional error from "learning from the data", which helps explore the positive effects of introducing LM after CoNN. However, we also proposed AutoCoNN, a method that automates the construction of CoNN using the LM's inherent code-writing and context-learning abilities. This allows the LM to generate CoNNs for specific tasks or domains with minimal need for human intervention.
>
>
> ---
>
> **Review:** Hard-coded policy of choosing the LM vs CoNN component by connecting that to task-based properties
>
> **Response:** We agree with your observation and see this as an exciting avenue for future research. In fact, for LM, the Neural Comprehension framework is plug-and-play without needing extra training. The hard-coded \beta avoids additional training. In our present work, we use a simple gating mechanism that switches between the LM and CoNN based on predefined rule-based tasks. The goal was to provide a first, proof-of-concept solution to demonstrate the potential benefits of combining LMs and CoNNs. As we discuss in the paper, a more sophisticated and learnable gating mechanism could be developed that adaptively decides whether to engage the LM or CoNN during the generation process. This improvement could lead to models capable of integrating and balancing text-based learning and rule-based learning more efficiently.
>
> ---
>
> **Review:** Authors claimed that their work suggests potential improvements from using Neural Comprehension in other tasks, but they did not mention how to get CoNNs for these tasks? In general, the discussion about CoNN design and implementation is somewhat skipped in the paper while it seem to be a crucial factor in this paper's impact.
>
> **Response:** Apologies for the oversight. We agree that CoNN design and implementation are crucial for our method. While we described the encoding of rules into CoNNs via Attention mechanisms (Select, Aggregate, and Zipmap) in Section 3.1 and provided further details in appendix B, we understand that this explanation might not have been sufficient. We will include more details on the CoNN design and implementation directly into the main part of the paper in the revision to ensure that the methodology is clearly understood.
>
>  Moreover, we developed a method named AutoCoNN (Appendix C) to construct CoNNs automatically for specific tasks using the code writing abilities of large language models, as detailed in Section number of AutoCoNN in the paper. Briefly, AutoCoNN employs a three-stage process: Observation, Induction, and Comprehension to automate the construction of CoNNs for various tasks and domains. We provide 24 different RASP examples that are used to generate CoNNs. With AutoCoNN, we only need to provide examples and two samples to generate a new AutoCoNN, we provided the relevant implementation in the code of the Supplementary Material. We will revise our manuscript to clarify this point.
>
> ```python
> from NeuralCom.AutoCoNN import AutoCoNN
>
> INSTRUCT = 'Create an SOp that is the last letter of a word'
> VOCAB = ['a','b','c','d','e','f','g']
> EXAMPLE = [[['a','b','c'],['c','c','c']],[['b','d'],['d','d']]]
>
> auto = AutoCoNN()
> model,tokenizer = auto(instruct=INSTRUCT,vocab=VOCAB,example=EXAMPLE)
> ```
>
> ---
>
> **Review:**  from my understanding you have done some hard coding of the beta policy such that i.e. it only triggers ICL gradient when it sees "=" token. Does this mean that this approach will not work at all if you use sequence "4 + 4 equals 8"? If so, do you have any ideas in mind how to make it work? In general I like this idea of plug-in CoNNs, but they need to be more seamless and fluent without requiring such hard coding in the task level.
>
> **Response:**  Thank you for your observation and important question. Indeed, in the current implementation of our model, we hard code to some extent when the CoNNs are triggered - especially when we use the "=" symbol as a trigger during the construction phase of CoNN. However, this is only one part of the more general problem of knowing when to apply a rule, which is an active area of research in itself. The "=" symbol was used as a clear, consistent, and identifiable marker to illustrate the capabilities of our approach.
>
> As for more complex cases where the operation symbols are phrased differently, as in "4 + 4 equals 8", the current model indeed would not trigger the CoNN. However, this does not mean our method is unable to handle such situations. We just need to also assign the equals method a meaning similar to "=" during the construction of CoNN, and perform the calculation. This is very easy for AutoCoNN. Or by additional training of the gating mechanism, our approach can also be extended to identify different operation wordings. For instance, we could train it to recognize different phrases that imply arithmetic operations and trigger the CoNN accordingly.
>
> The central aim of our work is to demonstrate the benefits of and propose a method for incorporating explicit rule knowledge into language models. The specific mechanisms by which these rules are triggered are largely an implementation detail. We agree with you that an ideal system would seamlessly integrate CoNNs and not require any hard coding, and we see the work presented in this paper as a first step towards this goal.

---

> > ### Comment · Reviewer_QFXM · 2023-08-16
> > **Thanks for your comments**
> >
> > Thanks for reviewers and authors for their feedback and discussion here. Indeed I did not read in every detail the appendix which describes AutoCoNN which made it harder to get relevant context.
> > In my opinion authors provided sufficient feedback and ensured some edits in the final manuscript. This types of work are quite different from mainstream LLM scaling/finetuning work and i think community will benefit from this types of research. I increased my score.

---

### Official Review · Reviewer_xyMG · 2023-07-05

**Soundness:** 4 excellent
**Presentation:** 4 excellent
**Contribution:** 4 excellent
**Rating:** 7
**Confidence:** 4

**Summary:**

While Large Language Models show promise for a wide swath of tasks, they are lacking when applied to symbolic reasoning tasks. To overcome this limitation, the authors propose to employ Compiled Neural Networks (CoNNs). They create networks specialised to arithmetic and symbolic tasks and propose a mechanism by which an LLM can propagate the gradient through CoNNs to better learn to solve symbolic reasoning tasks. They demonstrate improvement in pure symbolic manipulation (parity and reverse), arithmetic (addition and subtraction), and more complex symbolic reasoning (coin flip and last letter concatenation). They demonstrate better generalisation to out-of-distribution examples (proxied by digit length or sequence length for the first four tasks), a considerable improvement on LLC, and parity with a larger LLM when augmenting a smaller one (T5 small + NC v vanilla T5 large). The improvements are comparable to external tool-based approaches, however, hold a promise of better integration (as gradients can propagate without surrogates), and the mixture-of-experts style combination can be learnt rather than rule-based.

**Strengths:**

- By construction, CoNNs are interpretable from their basic building blocks, ensuring that paths that do go through them can be interpreted according to the rules they encode.
- The reduced number of parameters holds promise for reducing the cost of language models (relative to GPT-3)
- Even with simple gating, there is a non-trivial improvement on tasks when multiple CoNNs are employed (Section 5.4)

**Weaknesses:**

- There is an implicit assumption on practitioners to know what rules to expect and generate appropriate networks that then get used MoE style.
- Expanding on the previous, practitioners should be able to translate their rules, from, for example, regular expressions, to RASP to enable compilation to a NN.

**Questions:**

- While orthogonal to the paper, perhaps a remark on the difficulty of creating CoNNs appropriate for a task should be briefly discussed. For example, Section C.1 does discuss employing an LLM and ICL to obtain networks that approximate specific rules, how difficult is it to source examples for ICL? How many examples are needed before the output is of high enough quality? Can the correctness of the proposed CoNN be assessed by a non-expert? (as this last would be the use case, it is safe to assume an expert could write the network directly)

### Discussion Phase

The authors have clearly addressed the concerns raised during the rebuttal with clear and, as necessary, additional data. As long as such data gets exposed into the paper, via appendix, for example, I feel the paper has further improved.

**Limitations:**

The authors have address most limitations that arose as questions during the reading of the paper spare the one I listed under Questions with regards to the cost of producing CoNNs.

---

> ### Author Rebuttal · Authors · 2023-08-09
>
> Thank you for your positive evaluation and constructive feedback.
>
> In fact, as shown in the supplementary code, the process of AutoCoNN constructing CoNN requires Instruct (describing specific rules) and Example.
> ```python
> from NeuralCom.AutoCoNN import AutoCoNN
>
> INSTRUCT = 'Create an SOp that is the last letter of a word'
> VOCAB = ['a','b','c','d','e','f','g']
> EXAMPLE = [[['a','b','c'],['c','c','c']],[['b','d'],['d','d']]]
>
> auto = AutoCoNN()
> model,tokenizer = auto(instruct=INSTRUCT,vocab=VOCAB,example=EXAMPLE)
> ```
> ---
>
>
>  For Table 2 in the appendix, "Success by AutoCoNN" means using both Instruct and Example information. To resolve your confusion, we re-evaluated generating CoNN using only Instruct and only Example information separately in Table a. (Similarly, in the few-shot samples, only the corresponding information is used as well.)
>
> Table a:
> | CoNN Model | Expert's Working Time | Success by AutoCoNN | AutoCoNN (Instruct) | AutoCoNN (Example) |
> |-|-|-|-|-|
> | Parity Model  | 1 hour | 8/20 | 7/20 | 3/20 |
> | Reverse Model | 0.5 hour | 15/20 | 16/20 | 11/20 |
> | Last Letter Model | 0.5 hour | 13/20 | 12/20 | 10/20 |
> | Copy Model | 0.2 hour | 17/20 | 17/20 | 15/23 |
> | Addition Model | 48 hours | 0/20 | 0/20 | 0/20 |
> | Subtraction Model | 48 hours | 0/20 | 0/20 | 0/20 |
>
> We can easily find that, without predefining explicit rules but only "discovering" rules from examples, although the accuracy of AutoCoNN will slightly decrease, this scenario with only examples still has certain reliability.
>
>
> ---
>
> **Review:** While orthogonal to the paper, perhaps a remark on the difficulty of creating CoNNs appropriate for a task should be briefly discussed.
>
> **Response:** Thank you for pointing this out. We acknowledge that the description of the implementation details of AutoCoNN may not have been thorough enough. We will address your concerns in the following response and commit to revising the relevant content accordingly.
>
> ---
>
> **Review:** Section C.1 does discuss employing an LLM and ICL to obtain networks that approximate specific rules, how difficult is it to source examples for ICL?
>
> **Response:** The Supplementary Material `code\NeuralCom\AutoCoNN\prompts.py` contains 24 examples related to ICL, each consisting of Instruct, Example and Code. And here is an example:
>
> ```python
> def make_length() -> rasp.SOp:
>     """Creates the `length` SOp using selector width primitive.
>     Example usage:
>       length = make_length()
>       length("abcdefg")
>       >> [7.0, 7.0, 7.0, 7.0, 7.0, 7.0, 7.0]
>     Returns:
>       length: SOp mapping an input to a sequence, where every element
>         is the length of that sequence.
>     """
>     all_true_selector = rasp.Select(
>         rasp.indices, rasp.tokens, rasp.Comparison.TRUE).named(
>         "all_true_selector")  # Match tokens and tokens one by one, and calculate that they are completely equal.
>     return rasp.SelectorWidth(all_true_selector).named(
>         "length")  # which computes the number of elements in each row of a selector that with the value 1.
> ```
>
> These 24 examples are from Tracr's code [1]. We added the Instruct and Example in the comments for each example.
>
> ---
>
> **Review:** How many examples are needed before the output is of high enough quality?
>
> **Response:** Since RASP is a novel language, more examples are needed to meet the needs of ICL. In general, it is recommended to provide 16 or more samples, which can successfully generate CoNNs including Parity Model, Reverse Model, Last Letter Model, and Copy Model on the basis of GPT-3.5.
>
> ---
>
> **Review:** Can the correctness of the proposed CoNN be assessed by a non-expert?
>
> **Response:** Yes, it is possible to assess the correctness of CoNNs with just two examples. In practice, we first use GPT-3.5 to generate 20 different RASP codes, then convert them into CoNN models. We then test the 20 different CoNN models on two different examples sequentially. If they can get both examples correct, we can determine that the correct CoNN has been generated.
>
> This is because each layer of a CoNN is essentially an explicit transformation on the sequence. So a CoNN can either successfully transform all samples, or it is wrong.
>
> ---
>
> [1]Lindner D, Kramár J, Rahtz M, et al. Tracr: Compiled transformers as a laboratory for interpretability[J]. arXiv preprint arXiv:2301.05062, 2023

---

> > ### Comment · Reviewer_xyMG · 2023-08-13
> >
> > Thank you for the comments and for responding to my concerns!
> >
> > Table a is also interesting in that it shows that instruct and example are not purely additive/orthogonal as signals (although a Venn diagram of the solutions would be needed to better judge the situation, I feel the table provides better clarity).
> >
> > >  We then test the 20 different CoNN models on two different examples sequentially. If they can get both examples correct, we can determine that the correct CoNN has been generated.
> >
> > Is this restriction to two examples due to computational costs? It seems underpowered as a way to validate that a language check network has been correctly constructed.

---

> > > ### Author Response · Authors · 2023-08-15
> > >
> > > Thank you for your follow-up questions, which further clarify the points that need to be addressed about our work.
> > >
> > > ---
> > >
> > > **Regarding the non-additive nature of instructions and examples:**
> > > You're right; the instruct and example signals are not purely additive or orthogonal, but rather complement each other. The intersection of the two can provide more precise guidance for generating CoNNs. We agree that a Venn diagram could provide additional insight, and we will consider including such a visualization in the next version of the paper to better illustrate this relationship.
> > >
> > > **On the restriction to two examples for validation:**
> > > The most effective way to verify the correctness of CoNN is for experts to manually debug every line of RASP code to ensure correctness.
> > >
> > > For non-expert users who do not know the meaning of the RASP code, to facilitate demonstration and use of AutoCoNN, we suggest providing two examples to verify the correctness of CoNN. In Table c, we have tabulated the accuracy of 10 CoNNs as assessed manually by experts (us) after validation with different numbers of examples. It is easy to see that just 2 examples are sufficient to evaluate the correctness of CoNN.
> > >
> > > **Table c**
> > > | CoNN Model        | Example=1 | Example=2 | Example=5 |
> > > | ----------------- | --------- | --------- | --------- |
> > > | Parity Model      |     5/10      |    10/10       |   10/10        |
> > > | Reverse Model     |   10/10        |    10/10       |    10/10       |
> > > | Last Letter Model |   9/10        |     10/10      |      10/10     |
> > > | Copy Model        |     10/10      |      10/10     |      10/10     |
> > >
> > > The reason is that the process of converting RASP to CoNN is like runing Python code. If the CoNN can be generated normally, it is already half successful (most of the failed CoNNs in Table a above are due to code runing failures during conversion). After that, simple filtering can obtain a correct CoNN.
> > >
> > > ---
> > >
> > > Thank you again for your insightful questions and suggestions. They are instrumental in improving the clarity and thoroughness of our work.

---

> > > > ### Comment · Reviewer_xyMG · 2023-08-17
> > > >
> > > > Thank you for the clarification on the correctness check and Table c, I still feel that the number of examples needed will depend on the language validated, but for the tasks explored, I agree that n=2 looks sufficient.
> > > >
> > > > Thank you kindly for precisely addressing my concerns and the extra tables and example snippets.
> > > >
> > > > I will maintain my score at 7.

---

### Official Review · Reviewer_aik1 · 2023-07-09

**Soundness:** 2 fair
**Presentation:** 1 poor
**Contribution:** 3 good
**Rating:** 4
**Confidence:** 4

**Summary:**

The paper shows a strategy to improvise ICL by including CoNNs in the learning pipeline, which enable the LM to learn symbolic operations in addition to standard autoregressive LM generation. The resulting model is trained by a hand designed gradient accumulation technique and results are compared on symbolic tasks.

**Strengths:**

The paper shows how symbolic tasks can be included with general autoregressive training and therefore provides a way to train models following a few fixed symbolic tasks in mind. The paper provides a solid training recipe with proper mathematical justification. The results correlate with the claims and justify using the method.

**Weaknesses:**

It is unclear from the paper how different gating mechanisms are being derived in this network and how they are being included in the training framework. The authors say that \beta is not learned in the algorithm and essentially rule calculations are assigned to CoNNs. If that is true, the applicability seems a bit ad-hoc as different rules will then need to be hand written and not derived, and the benefits of the network will only be applicable to scenarios that are symbolically encoded. In other words, this seems to be a scalability challenge in terms of letting LMs learn rules. The experimental evaluation to justify the benefits seem very limited.

The paper also suffers from poor presentability with multiple grammatical errors, spelling mistakes etc. Also more context on training the overall CoNN+DNN system is missing from the paper or appendix.

**Questions:**

1. How does the method scale to new symbolic tasks?
2. How do we train a joint model and let the model learn the rules from data?

---

> ### Author Rebuttal · Authors · 2023-08-09
>
> We appreciate your detailed and constructive feedback very much. We hope this response helps address your concerns about the design and applicability of the method. We will update our documentation accordingly to make it more clear, and will consider your other suggestions.
>
> ---
>
> **Review:** It is unclear from the paper how different gating mechanisms are being derived in this network and how they are being included in the training framework.
>
> **Response:** We apologize for not elaborating on the gating mechanism in detail in our paper. The Neural Comprehension framework is a plug-and-play approach that does not require additional gradient training. The gating mechanism is simplified for most CoNNs as a rule-based predefined process: when we require a rule calculation, it triggers the gating variable to activate the CoNN.
>
> Since the focus of our research is on how to enable language models consisting solely of neural networks to have symbolic reasoning capabilities, the \beta determined by CoNN can avoid the extra errors brought by "learning from data". Determining the value of \beta during the construction of CoNN enables easy plug-and-play, such that built Addition model and Subtraction model can be readily used in models like GPT, T5, GLM-130B and Llama without training, thus enabling them to attain complete accuracy in performing addition and subtraction. Pre-determining \beta values provides greater adaptability compared to trainable \beta. We agree that a learnable gating mechanism could be more flexible; thus, we suggest implementing a learnable gating mechanism for future work involving larger-scale CoNN.
>
> ---
>
> **Review:**  this seems to be a scalability challenge in terms of letting LMs learn rules. The experimental evaluation to justify the benefits seem very limited.
>
> **Response:** Although currently we do need to manually write different rules for each CoNN, we propose the AutoCoNN toolkit in Appendix C. It utilizes the strong inductive abilities of LLMs like GPT-3.5 to automatically and pipeline-ly generate a large number of diverse CoNNs, which can be used in LMs. For example, for the four symbolic tasks in Figure 3, the CoNNs induced from the rule tasks can achieve full accuracy.
>
> From another perspective, thanks to the transferability of Neural Comprehension, these CoNNs from GPT-3.5/4 can enhance the performance of smaller language models such as GLM-130B in Figure 5. Although lacking the capability of calling external APIs (PAL methods), GLM-130B can also be enhanced in arithmetic capabilities by CoNNs.
>
> ---
>
> **Review:** The paper also suffers from poor presentability with multiple grammatical errors, spelling mistakes etc.
>
> **Response:** We apologize for the presentation errors; we'll ensure a thorough proofreading in the revised version to improve the paper's readability.
>
> ---
>
> **Review:** Also more context on training the overall CoNN+DNN system is missing from the paper or appendix.
>
> **Response:** Thank you for pointing this out. We admit that the description of the Language Models with CoNN may not be comprehensive enough. In the revision, we will add more details about the CoNN and LM systems, and clarify some parts that may be misleading, including:
> 1. Changing the layout of Figure 2 and adding caption explanations;
> 2. Moving some of the AutoCoNN content into the main text;
> 3. Adding a plug-and-play explanation for "CoNNs" in line 152.
>
> ---
>
> **Question:** How does the method scale to new symbolic tasks?
>
> **Response:** In the code section of the supplementary material, we provided an example of AutoCoNN generating a new CoNN: only "Instruct", "Vocab" and "Example" need to be provided, and the contextual learning capability of the LLM model can be utilized to generate a new CoNN model (We provide 24 examples about building CoNNs as few-shot prompts for AutoCoNN in advance.).
>
>
> ```python
> from NeuralCom.AutoCoNN import AutoCoNN
>
> INSTRUCT = 'Create an SOp that is the last letter of a word'
> VOCAB = ['a','b','c','d','e','f','g']
> EXAMPLE = [[['a','b','c'],['c','c','c']],[['b','d'],['d','d']]]
>
> auto = AutoCoNN()
> model,tokenizer = auto(instruct=INSTRUCT,vocab=VOCAB,example=EXAMPLE)
> ```
>
> ---
>
> **Review:** How do we train a joint model and let the model learn the rules from data?
>
> **Response:** At present, our work does not implement a mechanism for the model to directly learn rules from data. But just as we designed AutoCoNN, we suggest using LLMs to observe the data during the construction phase of CoNN and summarize its rules into a RASP language that can build CoNN.

---

> > ### Comment · Reviewer_aik1 · 2023-08-22
> > **Thanks for your comments**
> >
> > Thanks a lot for your comment, really appreciate your detailed descriptions. I do see the benefits and potential of this approach in adding symbolic knowledge to LLMs. This is a really interesting direction, and after reading the authors response in detail I am personally convinced this should be the direction to use symbolic constraints with LLM. The benefits of a plug and play model does have its benefits.
> >
> > I urge the authors to show the benefit of the approach with these symbolic modules on some public benchmark to showcase that the overall benefit does hold when any general purpose dataset is used, using multiple LMs if possible.
> >
> > I appreciate the hard work by the authors.

---

> > > ### Author Response · Authors · 2023-08-22
> > > **Responding to the reviewer's concerns regarding the Public Benchmarks**
> > >
> > > Thank you very much for your recognition.
> > >
> > > In the **original paper**, we selected public benchmarks from multiple perspectives.
> > >
> > > ---
> > >
> > > For **Symbolic Operation tasks (Section 5.1)**, we referred to [1] and [2], used similar experimental settings (i.e., set In-Dist and Out-of-Dist ranges), and likewise chose four types of symbolic operation tasks including *Parity*, *Reverse*, *Addition*, and *Subtraction*, performing comparisons among different scale models including *T5-small*, *T5-base*, *T5-Large*, *GPT-3*, *GPT-3.5*, and *GLM-130B*. And the Neural Comprehension achieved 100% accuracy.
> > >
> > > ---
> > >
> > > For **Symbolic Reasoning tasks (Section 5.2)**, we selected tasks identical to [3] (*Coin Flip* and *Last Letter Concatenation*), demonstrating the additional improvements brought by CoNN compared to the base models (*T5-small*, *T5-base*, *T5-large*).
> > >
> > > ---
> > >
> > > For **Arithmetic Reasoning tasks**,
> > >
> > > In **Section 5.3**, we selected the common AddSub dataset and manually expanded it to an arithmetic reasoning benchmark involving addition and subtraction reasoning of different digit numbers from 1 to 20 digits. On this benchmark, we chose three types of LMs: *GPT-3*, *GPT-3.5*, and *GLM-130B*. We also set up two baselines, the CoT baseline [3] and the PAL baseline that relies on an external API. The experimental results proved the benefits of our method over the baselines.
> > >
> > > In **Section Appendix D.2**, we conducted experiments with *GPT-3* and *GPT-3.5* on **five different real-world Arithmetic Reasoning public benchmarks** (these tasks are widely used to evaluate the reasoning capabilities of LMs [3][4]) including *GSM8K*, *SingleEq*, *AddSub*, *MultiArith*, and *SVAMP*. The method we proposed also demonstrated advantages in these benchmarks.
> > >
> > > ---
> > >
> > > **We believe that these public benchmark experiments from multiple angles are sufficient to demonstrate the overall benefits of Neural Comprehension. We hope that these information will help address your concerns and reconsider our ratings.**
> > >
> > >
> > > [1] Anil C, Wu Y, Andreassen A, et al. Exploring length generalization in large language models[J]. Advances in Neural Information Processing Systems, 2022, 35: 38546-38556.
> > >
> > > [2] Qian J, Wang H, Li Z, et al. Limitations of language models in arithmetic and symbolic induction[J]. arXiv preprint arXiv:2208.05051, 2022.
> > >
> > > [3] Wei J, Wang X, Schuurmans D, et al. Chain-of-thought prompting elicits reasoning in large language models[J]. Advances in Neural Information Processing Systems, 2022, 35: 24824-24837.
> > >
> > > [4] Kojima T, Gu S S, Reid M, et al. Large language models are zero-shot reasoners[J]. Advances in neural information processing systems, 2022, 35: 22199-22213.

---

### Author Rebuttal · Authors · 2023-08-09

Thank you to all the reviewers for their constructive feedback and recognition of our paper's contribution.

## Strengths
In the review, our paper was praised for its "**novel approach**", "**correct motivation**", "**high-quality experiments**", and "**superior performance**":
- We appreciate Reviewer aik1’s appraisal of `solid training recipe`, `proper mathematical justification` and `results correlate with the claims`.
- Furthermore, Reviewer xyMG acknowledged our approach in `demonstrate better generalisation to out-of-distribution examples`, `comparable to external tool-based approaches`, and `hold a promise of better integration`.
- They also recognized the high quality of our experimental investigation, a sentiment echoed by Reviewer QFXM who commended our for `The wide range of symbolic experimental tasks show that authors performed high quality experimental investigation`.
- Finally, Reviewer gBrw believes that our paper exhibits `arithmetic tasks (5.1) where the model achieves consistently 100% accuracy`, `The paper correctly identifies the limitations of LLMs`, and `the paper demonstrates strong results over competitive baselines on arithmetic/symbolic reasoning approaches`.

## Collective Concerns
The main criticisms centered on the issue of the hard-coded structure of CoNNs, how does the method scale to new symbolic tasks, and It lacks advantages compared to API methods.

**Hard-coded structue** We predefine \beta in the CoNN construction process, enabling the neural understanding framework to be used as a plug-and-play solution to enhance the symbolic capabilities of language models without requiring additional training. However, if \beta is a learnable parameter, pre-training of the language model is necessary, which adds to the cost of adoption. As we mentioned in our paper, we propose applying this method to larger-scale CoNNs.

**How does the method scale to new symbolic tasks** To facilitate the expansion of CoNN usage, we propose the AutoCoNN toolkit (see in Appendix C) in this paper for rapid CoNN construction. It only requires one instruction and two examples to generate a CoNN, making the extension of new CoNNs much easier.

**It lacks advantages compared to API methods** Figure 5 in the paper illustrates the arithmetic reasoning experiment. For LLMs like GPT-3.5, neural comprehension is no weaker than API methods. However, for weaker LMs like GLM-130B, due to their inability to invoke APIs, neural comprehension performance significantly outperforms API-based methods.


## Presentation
We have observed some constructive suggestions provided by the reviewers regarding the presentation. We commit to making rigorous revisions to the paper accordingly, as outlined below:
- Add a motivational explanation regarding Figure 1 in the Introduction;
- Changing the layout of Figure 2 (It's in the PDF file);
- Moving the AutoCoNN content into the main text;
- include an example of a CoNN in the Preliminaries section, such as demonstrating how to implement a Parity CoNN using Select, Aggregate, and Zipmap functions.
- Adding a plug-and-play explanation for "CoNNs" in line 152.
- line 47, `relying` -> `relies`
- line 50, `comprises` -> `comprise`
- line 136, `absolut` -> `absolute`
- line 336, `method` -> `methods`
- line 343 `facilitating the seamless` -> `facilitating seamless integration`


---

We believe that our efforts shown in this paper are a significant first stride towards perfecting Neural Comprehension. Your feedback has provided pointers to future works and we look forward to addressing these challenges in subsequent refinements of the system.

Thank you for your time and feedback, and for considering our research. We hope this rebuttal addresses the major concerns and makes a compelling case for the acceptance of our paper.

---

### Author Response · Authors · 2023-08-20
**Overall Response to Reviews**

**Thank you very much for reviewing our paper and offering insightful comments that help to improve our work. We hope we have addressed every reviewer's concerns, we look forward to your reply and thank you again for your time and effort.**

---

(**Concerns from reviewer gBrw**) We would briefly like to highlight what we believe is the main contribution of our work, which might not have been highlighted enough. It is difficult for pre-trained language models (PLM) to correctly perform symbolic tasks, further impacting their complex reasoning abilities, including arithmetic reasoning. The most recent work tend to address this issue by calling external API, but this naturally raises three new questions: Should language models rely on external interpreters? How can we reduce the computational resources used for generating extra code? What alternatives exist for language models that lack the capability to make API calls?

Our work proposes a novel framework that offers an alternative solution to address these three questions: The novel integration of CoNN and LM facilitates the combination of their complementary strengths in a plug-and-play fashion. It enhances the language model’s ability to handle rule-based questions. This approach is composed solely of neural networks and makes CoNN and PLM as a causal language model, Which makes it has more efficiency and high applicability compared to the method of calling APIs. The original paper's experiment of PAL (Figure 5) demonstrated that Neural Comprehension is not weaker than API-dependent methods. On the other hand, as stated by reviewer xyMG, "`The improvements are comparable to external tool-based approaches, however, hold a promise of better integration (as gradients can propagate without surrogates), and the mixture-of-experts style combination can be learnt rather than rule-based.`"

(**Concerns from reviewer aik1**) The rule calculation of \beta allows CoNN to be used in a plug-and-play fashion for language models, which is quite appealing. This type of integration, simple in nature, enables the language model to achieve a 100% accuracy rate without training. Moreover, we've described the proposed AutoCoNN toolkit in the original paper's appendices. The Parity Model, Reverse Model, Last Letter Model, etc., from the experimental section of the main text, have all been implemented using it, requiring only one instruction and two Examples to generate a new CoNN, thereby addressing the issue of CoNN's difficulty in expanding to other tasks. (**Reviewer QFXM has also been concerned with the rule calculations of \beta and the scalability of CoNN, but he acknowledged our rebuttal, stating "`authors provided sufficient feedback`".**)

---

We principally emphasize the scientific value of this unique framework to only utilize neural network to handle symbolic tasks. Historically, language models were far from mastering robust formal task, such as symbolic operations and arithmetic reasoning[1]. As commented by reviewer QFXM, "`This types of work are quite different from mainstream LLM scaling/finetuning work and i think community will benefit from this types of research.`"


[1] Stolfo A, Jin Z, Shridhar K, et al. A causal framework to quantify the robustness of mathematical reasoning with language models[J]. arXiv preprint arXiv:2210.12023, 2022.

---

### Decision · Program_Chairs · 2023-09-21

**Decision:**

Reject

**Comment:**

The paper overall presents a contribution that is possibly not highlighted enough, and important parts of it are delegated to the appendix rather than having a focus on them. There were also concerns about the scalability of the approach. Finally, some reviewers do not see the direct utility of adding symbolic support into the LLM the way the authors do. This paper could use further revision.